# Syntaxin-6, a Reliable Biomarker for Predicting the Prognosis of Patients with Cancer and the Effectiveness of Immunotherapy

**DOI:** 10.3390/cancers15010027

**Published:** 2022-12-21

**Authors:** Wenchao Li, Kuan Li, Hongfa Wei, Yu Sun, Yangjing Liao, Yuan Zou, Xiancong Chen, Cuncan Deng, Songyao Chen, Yulong He, Mingyu Huo, Changhua Zhang

**Affiliations:** 1Digestive Diseases Center, The Seventh Affiliated Hospital of Sun Yat-sen University, No. 628 Zhenyuan Road, Shenzhen 518107, China; 2Guangdong Provincial Key Laboratory of Digestive Cancer Research, The Seventh Affiliated Hospital of Sun Yat-sen University, No. 628 Zhenyuan Road, Shenzhen 518107, China; 3Department of Traditional Chinese Medicine, The Third Affiliated Hospital of Sun Yat-sen University, Guangzhou 510000, China; 4Department of Pathology, Southern Hospital, Southern Medical University, Guangzhou 510000, China

**Keywords:** syntaxin-6, pan-cancer, oncogene, immunotherapy

## Abstract

**Simple Summary:**

The treatment of cancer may differ on the basis of whether it is detected early or late, which determines whether the disease has spread, as well as treatment choices. Therefore, early detection is crucial in cancer therapy to increase overall patient survival and lessen the financial burden. Syntaxin-6 is a direct target of P53 and has been proven to be an oncogene in several types of cancer. In this research, we comprehensively analyzed the carcinogenic role of STX6 in pan-cancer, and we found that STX6 may also play an essential role in the tumor microenvironment, and that knocking down STX6 could enhance the effect of anti-PD-1.

**Abstract:**

Syntaxin-6 (STX6), a vesicular transport protein, is a direct target of the tumor suppressor gene P53, supporting cancer growth dependent on P53. However, STX6’s function in the tumor microenvironment has yet to be reported. In this research, we comprehensively explored the role of the oncogene STX6 in pan-cancer by combining data from several databases, including the Cancer Genome Atlas, CPTAC, cBioPortal, and TIMER. Then, we verified the carcinogenic effect of STX6 in hepatocellular carcinoma (HCC) and colorectal cancer (CRC) through a series of experiments in vitro and in vivo. Bioinformatics analysis demonstrated that STX6 is an oncogene for several cancers and is mainly involved in the cell cycle, epithelial–mesenchymal transition, oxidative phosphorylation, and tumor immune modulation, especially for tumor-associated fibroblasts (CAFs) and NKT cells. Additionally, a high level of STX6 could indicate patients’ resistance to immunotherapy. Our own data indicated that the STX6 level was upregulated in HCC and CRC. Knockdown of the STX6 levels could arrest the cell cycle and restrain cell proliferation, migration, and invasion. RNA-seq indicated that STX6 was significantly involved in pathways for cancer, such as the MAPK signal pathway. In a mouse model, knockdown of STX6 inhibited tumor growth and potentiated anti-PD-1 efficacy. In light of the essential roles STX6 plays in carcinogenesis and cancer immunology, it has the potential to be a predictive biomarker and a target for cancer immunotherapy.

## 1. Introduction

Cancer is a leading cause of death and a significant barrier to increasing life expectancy [1]. According to the World Health Organization (WHO), cancer was the leading cause of death for those at age of 30–70 in 112 of 183 countries in 2019. It ranked third or fourth in another 23 countries [2]. With the increase in the elderly population around the world, the risk of cancer from age-related health deterioration is constantly increasing [3]. Despite the limitations and adverse effects that come with cancer therapy, it has achieved considerable technical improvements over the last century [4]. The three mainstays of cancer care are surgery, radiotherapy, and chemotherapy. However, immunotherapy has emerged as the fourth central pillar in the battle against illness, using immune checkpoint inhibitors, chimeric antigen receptor (CAR) T-cell therapy, and cancer vaccines to treat cancers by harnessing the body’s immune system to detect and target cancer cells [5]. Cancer treatment may differ on the basis of whether it is detected early or late, determining whether the disease has spread, as well as treatment choices. Indeed, early detection is crucial in cancer therapy to increase overall patient survival and lessen the financial burden [6]. Consequently, discovering more sensitive cancer biomarkers and enhancing the efficiency of tumor immunotherapy are urgent needs.

Syntaxin-6 is a vesicular transport protein, mainly located in the Golgi and endosomal membranes, and it is responsible for the intracellular transport of specific proteins to maintain the homeostasis of their generation and degradation [7,8]. STX6-dependent trafficking is vital for diverse cellular functions, including developmental and pathological processes. Previous studies have proven that STX6-mediated intracellular transport of vascular endothelial growth factor 2 (VEGFR2) [9], epidermal growth factor receptor (EGFR) [10], and integrins [11] regulates tumor proliferation, angiogenesis, and metastasis. In addition, as a vesicle transport protein, STX6 can participate not only in the regulation of the size and number of extracellular vesicles [12], but also in the efflux of cancer cells to chemotherapy drugs through the form of vesicle transport to promote cell resistance to chemotherapy [13,14]. Recent studies have demonstrated the carcinogenic role of STX6 in many types of cancer, including cervical cancer [15], renal cell carcinoma [16], and pancreatic ductal adenocarcinoma [17]. Moreover, STX6 has been proven to be a direct target of P53 and is required for cell adhesion and survival [18], reinforcing the notion that STX6 plays an oncogenic role in tumors. STX6 is involved in the uptake and excretion of inflammatory granules [19,20] and TNF-a [21] from neutrophils and macrophages, respectively, in immune cells. Therefore, STX6 may also play an essential role in the tumor microenvironment. In this study, we comprehensively analyzed the carcinogenic role of STX6 in pan-cancer and explored whether it also plays an important role in the tumor microenvironment.

## 2. Materials and Methods

### 2.1. Collection and Analysis of STX6 Transcriptome and Proteomic Sequencing Data

Downloads of TCGA (https://portal.gdc.cancer.gov/, accessed on 17 January 2022) and GTEx (https://gtexportal.org/home/, accessed on 17 January 2022) uniformly normalized transcriptome datasets (PANCANCER, N = 19,131, G = 60,499) and proteomics data came from the CPTAC database (https://cptac-data-portal.georgetown.edu/, accessed on 17 January 2022). For the log scale, the transformation Log2(TPM + 1) was utilized. Cancer types with fewer than three samples were excluded. Finally, we collected the proteome expression data of 10 distinct cancer tissues, as well as the transcriptome expression data of 33 different cancer tissues and 32 different normal tissues. Calculations were performed using R software (version 3.6.4) to compare the expression levels of normal and tumor samples. Statistically significant differences were determined to be * *p* < 0.05, ** *p* < 0.01, and *** *p* < 0.001 according to *t*-tests on these tumor types.

### 2.2. STX6 Genetic Alteration Analysis

The 32 cancer types of TCGA were examined for STX6 genetic changes using the cBioPortal database (www.cbioportal.org, accessed on 17 January 2022) [22]. Among the components of genomic modifications are mutations, structural variations, amplifications, deep deletions, and multiple alterations. The frequencies of STX6 copy number variations and mutations were noted in all TCGA tumors, and the data are shown using plotted bar charts. Additionally, data on each tumor’s HM450 and HM270 methylation were gathered from the cBioPortal database.

### 2.3. Survival Analysis

The R (survival) package (https://cran.r-project.org/mirrors.html, accessed on 20 January 2022) was used to compute the Kaplan–Meier plotter and log-rank test for the survival curves of patients from TCGA database. To ascertain if STX6 was linked to survival outcomes, including overall survival (OS), disease-specific survival (DSS), progression-free interval (PFI), and disease-free interval (DFI), Cox proportional hazards models, predicting the hazard ratio, were built. Univariate survival analysis was used to generate the hazard ratio (HR), 95% confidence intervals, and log-rank *p*-value; a *p*-value of 0.05 or below was deemed statistically significant. The “RMS” program was used to build the nomogram integrating the expression of STX6 and clinicopathological risk variables, and the concordance index was used to evaluate it (C-index) quantitatively.

### 2.4. Immunohistochemistry

Immunohistochemical staining on eight common malignancies was chosen to experimentally confirm the protein levels of STX6 in tumor and healthy tissues. All patients who gave informed permission were used to gather the samples. The Seventh Affiliated Hospital of Sun Yat-sen University’s internal review and ethical committees gave their consent for the samples used in this investigation. After being embedded in paraffin and treated with formalin, normal and tumor tissues were then sliced into 5 µm thick slices using a microtome. The sections were dewaxed in xylene, rehydrated in graded alcohols to distilled water, and then placed into EDTA for boiling to expose the antigen before being incubated with hydrogen peroxide to lower endogenous peroxidase activity. The sections were treated with rabbit anti-STX6 antibody (Abcam, #ab140607, IHC:1:1000) at 4 °C overnight after being blocked with 5% goat serum. The slices were then treated with a secondary goat anti-rabbit IgG biotinylated antibody (ZSGB-BIO, #PV-6001) for 30 min at room temperature. The 3,5-diaminobenzidine (DAB) substrate kit was used to view the sections, which caused a brown precipitate to form at the antigen location. Hematoxylin was then used as a counterstain for 2 min. Using a semi-quantitative method, the staining intensity was graded as follows: 0, unfavorable; 1, frail; 2, moderate; 3, powerful. These percentages were used to identify the frequency of positive cells: 0, less than 5%; 1, 5–25%; 2, 26–50%; 3, 51–75%; 4, more than 75%. By multiplying the staining intensity and the frequency of positive cells, the final IHC scores were calculated (score = (number of pixels in a zone) × (score of the zone)/(total number of pixels in the image)). When the tissue staining was uneven, each region was rated separately, and the total of the individual scores was the outcome.

### 2.5. Immune Infiltration Analysis

The TIMER2.0 database (http://timer.comp-genomics.org/, accessed on 2 February 2022) can be used to evaluate the relationship between immune cell infiltration and STX6 expression, and to systematically study the levels of immune infiltrates in pan-cancer as determined by several algorithms. The main focus of this work was the association between STX6 expression and tumor stromal cells, such as cancer-associated fibroblasts, endothelial cells, and natural killer T cells. Spearman’s correlation based on tumor purity correction was performed to conduct the association study.

We received a globally standardized pan-cancer dataset from the UCSC (https://xenabrowser.net/, accessed on 2 February 2022) database. The “estimate” R program and Spearman’s correlation analysis were used to determine each tumor’s STX6 expression, ImmuneScore, and StromalScore. By using the flag genes for the various immune cell types, single-sample gene set enrichment analysis (ssGSEA) was used to determine immune cell infiltration [23,24]. The correlation between STX6 and 24 different types of immune cell infiltration in the tumor samples from THYM, DLBC, LGG, UVM, UCEC, and KIRC was examined using a lollipop chart and Spearman correlation analysis. Meanwhile, a vioplot was created to analyze the Wilcoxon rank-sum test (*p* < 0.05, statistically significant) to determine the association between STX6 and the recruitment of immune cells. Patients were split into two groups (high and low STX6 expression based on the median STX6 expression level) for each TCGA tumor type in order to compare the degree of immune cell infiltration.

### 2.6. Immunotherapy Prediction Analysis

The effectiveness of PD-1/PD-L1 inhibitors is significantly connected with the tumor mutation burden (TMB). Most clinical trials employing TMB as a metric have successfully achieved the end goal, with almost no failure. We determined the TMB scores for each TCGA sample. A disorder known as microsatellite instability (MSI) is characterized by repeating mono- and oligonucleotide sequences (short tandem repeats) that are indicative of a defect in DNA mismatch repair (MMR). A crucial clinical tumor marker is microsatellite instability (MSI) coupled with DNA mismatch repair deficiencies. Spearman’s correlation coefficient was used to assess the association between STX6 expression and TMB or MSI.

The ROC plotter online database (https://www.rocplot.org/, accessed on 5 February 2022) uses transcriptome-level data from patients with breast, ovarian, and colorectal cancer, as well as glioblastomas, to correlate gene expression with therapeutic response. Through this online database, we investigated the relationship between STX6 and the effectiveness of immunotherapy in this research.

### 2.7. Gene Set Enrichment Analysis

In order to determine the normalized enrichment score (NES) and false discovery rate (FDR) of the DEGs between the low- and high-STX6-expression cancer groups for each biological process in each cancer type, the “gmt” file of the hallmark gene set (h.all.v7.5.1.symbols.gmt), which contains 50 hallmark gene sets, was downloaded from the website of the Molecular Signatures Database (MSigDB, https://www.gsea/index.jsp, accessed on 11 June 2022). GSEA was carried out using the R program “clusterProfiler”, and the bubble plot generated using the R package “ggplot2” represents the findings in a concise manner.

### 2.8. Western Blotting

The BCA protein assay kit (Pierce) was used to measure CRC or HCC cell line protein concentrations (CRC: NCM-460, HCT15, SW480, HCT116, and HT-29; HCC: L02, HepG2, Huh7, BEL-7404, MHCC97H, SMMC7721, and QGY-701). An amount of 20 µg of protein was isolated on 10% SDS poly acrylamide gels and transferred to PDVF membranes. Membranes were blocked with 5% BSA diluted in Tris-buffered saline containing Tween-20 (TBS-T) for 60 min and then incubated with anti-STX6 primary antibody (1:1000 dilution, Abcam, ab140607) at 4 °C overnight, followed by goat anti-rabbit IgG secondary antibody (1:5000 dilution) at 37 °C for 1 h. β-Actin (1:4000 dilution, CST, #3700) was utilized as a loading control for immunodetection (ECL, Guangzhou, China).

### 2.9. Quantitative Reverse Transcription-PCR (qRT-PCR)

Total RNA from CRC or HCC specimens was extracted using Trizol (Life Technologies, CA, USA) and reverse-transcribed with the Transcriptor cDNA Synth Kit (Roche, Shanghai, China). qRT-PCR was conducted using SYBR-green PCR Master Mix and 45 cycles of 95 °C for 10 s, 60 °C for 20 s, and 72 °C for 20 s.

### 2.10. Cell Culture and Plasmid Transfection

HCC and CRC cell lines were cultured in DMEM or RPMI-1640 with 10% fetal bovine serum and incubated in a humidified incubator with 5% CO_2_ at 37 °C. All cell lines were obtained from Yulong He (Sun Yat-sen University, Shenzhen, Guangdong, China). The STX6 full-length plasmid was transfected using Lipofectamine 3000 (Invitrogen, California, USA). GeneCopoeia was used to produce the STX6 and control plasmids (GeneCopoeia, Guangzhou, China). These sequences are defined as follows:

sh1: forward, 5′–TAATACGACTCACTATAGGG–3′; reverse, 5′–CTGGAATAGCTCAGAGGC–3′.

sh2: forward, 5′–TAATACGACTCACTATAGGG–3′; reverse, 5′–CTGGAATAGCTCAGAGGC–3′.

Sh3: forward, 5′–TAATACGACTCACTATAGGG–3′; reverse, 5′–CTGGAATAGCTCAGAGGC–3′.

Negative control: forward, 5′–GCGGTAGGCGTGTACGGT–3′; reverse, 5′–ATTGTGGATGAATACTGCC–3′.

OE: forward, 5′–GCGGTAGGCGTGTACGGT–3′; reverse, 5′–ATTGTGGATGAATACTGCC–3′.

### 2.11. EDU Cell Proliferation and Cell Cycle Assays

HCC and CRC cells were fixed with 4% paraformaldehyde for 20 min, washed three times for 3 to 5 min each with PBS, and then subjected to EDU tests. The remaining procedures were conducted using the instructions provided in the EDU kit, and fluorescence was evaluated using a microscope. The cell cycle was determined by flow cytometry after tumor cell lines were collected with 0.25% trypsin, resuspended in PBS, incrementally added to absolute ethanol, cultured for 12–24 h, and finally stained as instructed.

### 2.12. Migration and Invasion Assays

A total of 3 × 10^4^ serum-free cells were planted in the top chamber of the Transwell invasion system (Corning Incorporated Costar, Corning, USA), and 600 µL of DMEM or 1640 containing 10% FBS was added to the bottom chamber to generate a serum concentration gradient. The number of cells that passed through an 8 µm polycarbonate membrane after being incubated at 37 °C in a humidified incubator with 5% CO_2_ was determined. Invasion assays followed the same protocols as the migration assays, but Matrigel was precoated in the top chamber for 2 h before seeding cells.

### 2.13. CRC Mouse Model

The institutional animal care and use committee (IACUC) of the Seventh Affiliated Hospital of Sun Yat-sen University authorized the experimental methods and animal use and care protocols. Three week old male BALB/c mice (n = 8 each group) were obtained from Shenzhen topBiotech company (Shenzhen, Guangdong, China). Each mouse’s right flank received 5 × 10^5^ CT26 cells. The tumor volume was determined every 4 days using the method 0.5 × length × width. One week after the cells were inoculated, anti-mouse PD-1 (Bio-X, #RMP1-14) was injected every 5 days.

### 2.14. Statistical Analysis

The Wilcoxon rank-sum test was used to determine statistical significance to evaluate STX6 expression levels between normal tissues and malignant tissues. The statistical significance of the levels of STX6 protein expression in clinical HCC and CRC samples and surrounding tissues was assessed using a paired *t*-test. The Kaplan–Meier technique and univariate Cox regression analysis were used to evaluate the predictive value of STX6 expression in each malignancy. The statistical correlations between STX6 and many other parameters were evaluated using Spearman correlation analysis. Using the chi-square test, the statistical significance was calculated for the proportions of low- and high-STX6 cancer groupings that responded to ICI treatment and those that did not.

## 3. Results

### 3.1. STX6 Expression Levels in Pan-Cancer

STX6 mRNA expression levels were examined between tumor and surrounding normal tissues in 32 cancer types using data from TCGA and GTEx. There was an increase in the mRNA expression of STX6 in tumor tissues compared to neighboring normal tissues in all cancers (Figure 1A,B). With regard to STX6 protein levels, data from the CPTAC database indicate that STX6 was overexpressed in cancer tissues such as clear-cell RCC, UCEC, lung cancer, pancreatic cancer, head and neck cancer, and liver cancer, but not in GBM (Figure 1C).

### 3.2. Genetic Alteration of STX6

We investigated the STX6 alteration frequency and mutation count in cancer patients using the cBioPortal database. Overall, cancers with STX6 mutations were predominantly amplified mutations, notably cholangiocarcinoma (>8%), liver hepatocellular carcinoma (>8%), and breast invasive carcinoma (>7%) (Appendix A). DNA methylation plays a vital function in transcriptional regulation and may lead to the silencing or inactivation of tumor suppressor genes, thus promoting the start and proliferation of malignancies [25]. In this work, we discovered that the expression level of STX6 was inversely linked with the amount of promoter methylation in 25 types of cancer we examined. The six greatest negative relationships (BRCA (r = −0.34, *p* = 2.37 × 10^−27^), ESCA (r = −0.37, *p* = 3.46 × 10^−7^), DLBC (r = −0.38, *p* = 0.019), MESO (r = −0.41, *p* = 1.37 × 10^−4^), SKCM (r = −0.41, *p* = 2.73 × 10^−16^), and ACC (r = −0.43, *p* = 1.08 × 10^−4^)) are presented in Appendix A. Furthermore, in the association study between STX6 and copy number variation (CNV), another indication demonstrates the deletion or amplification of genomic DNA in the full chromosomal set of cancer and genetic illnesses. For the tumor, the deletion fragment may include tumor suppressor genes, whereas the amplified fragment may contain oncogenes [26]. The findings show that, in 29 different types of cancer, STX6 mRNA expressions were mostly positively linked with CNV. Appendix A display the six strongest positive correlations: UCS (r = 0.58, *p* = 3.255 × 10^−6^), LUAD (r = 0.59, *p* = 3.39 × 10^−48^), MESO (r = 0.59, *p* = 3.89 × 10^−9^), SKCM (r = 0.61, *p* = 5.1 × 10^−39^), PAAD (r = 0.58, *p* = 2.67 × 10^−19^), and BRCA (r = 0.58, *p* = 2.01 × 10^−128^).

### 3.3. The Prognostic Value of STX6 mRNA in Pan-Cancer

STX6 was strongly related to the survival probability of a total of 14 cancer types (Appendix A), as shown by the Kaplan–Meier curves. STX6 was a poor predictor of patient survival in ACC (*p* = 0.005; HR = 17.33), BRCA (*p* = 0.012; HR = 1.55), CESC (*p* = 0.019; HR = 1.78), HNSC (*p* = 0.022; HR = 1.39), KIRP (*p* < 0.001; HR = 5.18), LIHC (*p* = 0.001; HR = 1.82), MESO (*p* = 0.028; HR = 1.95), and OSCC (*p* = 0.038; HR = 1.44) among these malignancies (Appendix A). Meanwhile, STX6 was a favorable predictor of overall survival for patient with ESAD (*p* = 0.008; HR = 0.42), ESCA (*p* = 0.011; HR = 0.51), GBMLGG (*p* = 0.001; HR = 0.61), LAML (*p* = 0.027; HR = 0.58), OV (*p* = 0.011; HR = 0.68), and UCEC (*p* = 0.029; HR = 0.64) (Appendix A).

An investigation using the Cox regression model was carried out to validate whether or not STX6 was connected to the risk of survival. Figure 2 shows a positive correlation between STX6 and the hazard ratios of DFI, DSS, OS, and PFI in ACC, LIHC, PAAD, and KIRP. On the other hand, a negative association existed between STX6 and these parameters in OV, which suggests that STX6 is a more likely risk factor for patients who have cancer.

STX6’s predictive relevance in malignancies drove us to further study STX6-detrimental and STX6-favorable cancer subtypes. Combining clinical parameters with STX6 expression led to the development of a nomogram that accurately predicts the prognosis of cancer patients. Overall, the nomogram points demonstrated that patients with ACC, KIRP, LIHC or PAAD had shorter 5 year survival rates when STX6 expression was high (Appendix A), but OV showed the opposite (Appendix A).

### 3.4. GSEA of STX6 Analysis

To study the biological role of STX6 expression in LIHC, KIRP, and OV, we initially conducted GSEA of STX6 analysis on KEGG pathways. The findings revealed that STX6 was favorably related to the cell cycle, MAPK signaling pathway, TGF BETA signaling pathway, VEGF signaling circuit, and cancer pathways in ACC, KIRP, LIHC, or PAAD (Appendix A), but OV showed the contrary (Appendix A).

Cancer-associated fibroblasts and endothelial cells are tightly associated with the TGF BETA and VEGF signaling pathways, indicating that STX6 is likely implicated in the tumor microenvironment and cancer metastasis. We next thoroughly explored the biological importance of STX6 in cancer hallmark gene sets. GSEA was used to identify STX6-associated cancer hallmarks by comparing gene expression differences between low-STX6 and high-STX6 subgroups in each malignancy. The median value of mRNA expression was taken as the threshold for each cancer type. It was found that STX6 expression was linked to various immunological pathways, including TNFA signaling via NFKB, IFN-α, IFN-β, and inflammation, as well as allograft-rejection pathways in kidney cancer (KICH, KIRC, and KIRP), CESC, and PAAD; however, this was not the case for other cancers, including LGG, MESO, OV, SARC, SKCM, and STAD (Figure 3). Furthermore, STX6 expression was favorably connected to the mitotic spindle, G2M checkpoint, and epithelial–mesenchymal transition, demonstrating that STX6 was strongly related to tumor proliferation and metastasis.

### 3.5. Tumor Microenvironment and Immune Cell Infiltration Analyses

We anticipated that tumor-associated fibroblasts and vascular endothelial cells might be closely connected to STX6 based on the KEGG findings for STX6 in malignancies. The TIMER database was used to support this claim. According to the data shown in Appendix A, STX6 was positively connected with tumor-associated fibroblasts in 22 of the 40 cancer tissues studied (EPIC, MCPCOUNTER, and TIDE). Furthermore, endothelial cells had a favorable correlation with STX6 in most cancers (18 of 40 cancer types).

As part of the tumor microenvironment, tumor-associated fibroblasts and vascular endothelial cells play a role in tumor growth, invasion, and metastasis. Herein, we explored the associations between STX6 expression and tumor microenvironment makeup by adopting the ESTIMATE method to compute the immune and stromal scores. Among the 32 cancer species studied, five of them (COAD, LAML, KIRP, and KIRC) were found to have a significant positive connection with stromal scores; the other 10 (GBM, LGG, BRCA, STES, STAD, UCEC, THYM, THCA, and BLCA) were found to have a significant negative association with STX6 expression (Appendix A). As for immunological scores, four had substantial positive connections (COAD, KIRC UVM, and DLBC), whereas 14 had significant negative connections (GBM, LGG, BRCA, ESCA, STES, SARC, STAD, UCEC, HNSC, LUSC, THYM, THCA, SKACM, and OV) (Appendix A). More details are displayed in Table 1.

Given the stromal and immunological scores of STX6 in pan-cancer, tumors (THYM, DLBC, LGG, UVM, UCEC, and KIRC) having a strong association between STX6 and immune/stromal scores were chosen to analyze the enrichment score of STX6 in each immune cell type using single-sample gene set enrichment analysis (ssGSEA) [23]. The lollipop chart demonstrated that malignant tissues with greater STX6 levels exhibited a larger percentage of TCM cells and T helper cells, but lower NK cell levels (Figure 4A–F). Furthermore, the relationship between STX6 levels and NKT cells was exported from the TIMER database. The results showed that STX6 was significantly negatively correlated with NKT cells in pan-cancer, which suggests that STX6 may help immune escape by stopping the expression of NKT cells (Figure 4G).

### 3.6. The Predictive Role of STX6 in Cancer Immunotherapy

TMB and MSI have long been used to predict immunotherapy’s efficacy in various cancers [27]. In this study, we proceeded to explore the link between STX6 and tumor TMB and MSI in pan-cancer, as illustrated in Figure 5. The link between STX6 expression and TMB gained significance (*p* < 0.05) in eight types of cancer. In general, STX6 expression was favorably connected with TBM in ACC, LGG, LAML, SKCM, STAD, and PAAD while being negatively correlated in THCA and ESCA (Figure 5A). Additionally, we discovered that the expression of STX6 showed a favorable link with READ, LAML, COAD, LUSC, and UCEC, but a negative relationship with the MSI of five different cancers, including DLBC, LGG, HNSC, THCA, and SKCM (Figure 5B).

The results from the studies above indicate that STX6 could predict the efficacy of immunotherapy (ICI) in the malignancies studied. Anti-PD-L1 and anti-PD-1 antibodies have made a significant contribution to the immunotherapy of malignancies, and their use is expected to grow in the future [28]. We download the tumor sequencing data linked to PD-L1 or PD-1 therapy from the GEO database and homogenized the data. Prostate cancer, colorectal cancer, bladder cancer, melanoma, and urothelial cancer were collected in the PD-1 group, while those collected in the PDL-1 group were ureter/renal pelvis and bladder cancer, esophageal adenocarcinoma, and retinoblastoma. The PD-1 and PDL-1 treatment groups were compared regarding tumors included in both groups. In the nonresponse group, STX6 mRNA expression was greater than in the response group independent of treatment with PD-L1 or PD-1 (Figure 5C,D).

### 3.7. STX6 Level Was Regulated in Multiple Cancer Clinical Specimens

To further corroborate the data provided by our bioinformatics, we conducted immunohistochemical analysis to evaluate the protein levels of STX6 in tumor and normal tissues in common tumor types, including THCA, ESCA, STAD, COAD, PAAD, LUSC, and KRC. As expected, the STX6 staining score in tumor tissues was higher than that in normal tissues in all cancer types included in the study (Figure 6).

In order to further confirm STX6’s ability to cause cancer and its oncogenic mechanism, we chose to focus on the HCC and CRC cancer types based on the findings of the aforementioned analysis of STX6’s role in pan-cancer. In HCC and CRC cancer tissues, as shown in Figure 7, STX6 mRNA was expressed at a higher level than in normal tissues (Figure 7A,B), and the same pattern was seen at the cell line level. STX6 protein expression was elevated in the HCC (Figure 7C) and CRC (Figure 7D) cancer cell lines.

### 3.8. STX6 Knockdown Causes Cell Cycle and Metastasis Halt

To explore its biological function and mechanism, we established the STX6 knockdown cancer cell lines BEL-7404 and HCT-116 by lentivirus transfection. EDU proliferation detection showed that STX6 knockdown inhibited BEL-7404 and HCT116 cell growth (Figure 7E). Flow cytometry analysis indicated that STX6 knockdown could stop cell-cycle progression in the G2/M phase (Figure 7F). Furthermore, the migration and invasion abilities of cancer cells were weakened when the STX6 level was decreased (Figure 7G,H).

### 3.9. Silencing STX6 Inhibits Tumor Growth and Potentiates Anti-PD-1 Efficacy In Vivo

RNA-seq of BEL-7404 cells with STX6 knockdown was then conducted to further investigate the molecular mechanism of STX6 participation in carcinogenesis. STX6 was primarily implicated in tumor growth through the microRNA, MAPK, and TNF signaling pathways (Figure 8A). By extracting cell proteins with different expression levels of STX6, it was found that knocking down the STX6 level could inhibit the expression of EMT-associated proteins, as well as the phosphorylation levels of MRK and ERK proteins, but overexpression of the STX6 level showed the opposite result (Figure 8B,C).

To further evaluate STX6’s critical involvement in carcinogenesis and progression, we established a xenograft cancer model by injecting CT26 cells expressing varying quantities of STX6 into Balb/c mice. We discovered that the STX6 knockdown group’s tumors were lower in volume and weight than the control group (Figure 8D–F). In view of the results above suggesting that STX6 expression level can predict whether patients are resistant to immunotherapy, and the transcriptional RNA_sequencing results suggested that the expression of PD-L1 in BEL-7404 cells was up-regulated after silencing the level of STX6 (data not shown). We continue to explore whether knockdown of STX6 can enhance the efficacy of immunotherapy (schematic diagram shown in Figure 8G). As we expected, the knockdown of STX6 could potentiate the anti-PD-1 efficacy (Figure 8H–J). Analyses of removed tumors using immunofluorescence (immunohistochemistry) revealed that STX6 knockdown significantly increases the infiltration level of CD8α in the tumor microenvironment (Figure 8K,L).

## 4. Discussion

Studies showing syntaxin-6’s crucial function in cellular processes have also highlighted the significance of syntaxin-6-dependent trafficking of several human disease-associated proteins.

In granulocytes, STX6 aids in the exocytosis of inflammatory granules and cytokines [29]. STX6 and SNAP-23 promote gelatinase granule (GG) and specific granule (SG) secretion in active neutrophils [20], while STX6 and the Vti1b complex hasten TNF production in activated macrophages [21]. When it comes to endothelial cells, stx6 controls the trafficking of VEGFR2 from the trans-Golgi network (TGN) to the plasma membrane [9]. Additionally, it controls the recycling of integrin 51, which is involved in interacting with the fibronectin found in the extracellular matrix [30]. Interference in syntaxin-6 function results in degradation of VEGFR2 and integrin 51 and a failure of angiogenesis, both of which rely on trafficking via TGN and early endosomes (EEs) to regulate their functions.

Many studies have highlighted STX6 as a possible therapeutic target in cancer. By encouraging cell-cycle progression, cell metastasis, and treatment resistance, STX6 may facilitate tumor progression. In our work, we utilized a systematic pan-cancer analysis and in vitro and in vivo experiments to determine that STX6 expression is increased in most cancers and may be used as an independent risk factor to anticipate patient survival. In addition, STX6 may have a role in a number of other biological processes that contribute to tumor development and progression. These findings further reaffirm the promise of STX6 as a prognostic marker. Moreover, we discovered that STX6 might enhance tumor growth via involvement in CAFs and T-cell infiltration in the tumor microenvironment. With this new evidence, we may go on with our investigations into the molecular mechanism, especially the MAPK/ERK pathway, via which STX6 contributes to carcinogenesis and progression with more certainty.

To the best of our knowledge, this is the first report of a thorough examination of STX6 in pan-cancer, and in vitro and in vivo studies were used to investigate STX6’s oncogenic mechanism in HCC and CRC. However, some limitations also apply to this study. Firstly, we only confirmed the connection between STX6 and cell-cycle progression and metastasis in liver cancer and colorectal cancer, leaving out the verification of other biological processes including apoptosis or drug resistance. Secondly, we only created colorectal cancer animal models; there were no liver cancer animal models for in vivo investigations. Lastly, this work did not investigate the precise method via which STX6 is involved in the tumor microenvironment. Future research should aim to use transgenic animal models to better investigate the precise molecular mechanism of STX6 participation in carcinogenesis and development.

## 5. Conclusions

In summary, our work discovered that STX6 may control T-cell infiltration in the tumor microenvironment, and that it has an oncogenic function in a number of malignancies. STX6 is a reliable biomarker for predicting the prognosis of patients with cancer and the effectiveness of immunotherapy.

## Figures and Tables

**Figure 1 cancers-15-00027-f001:**
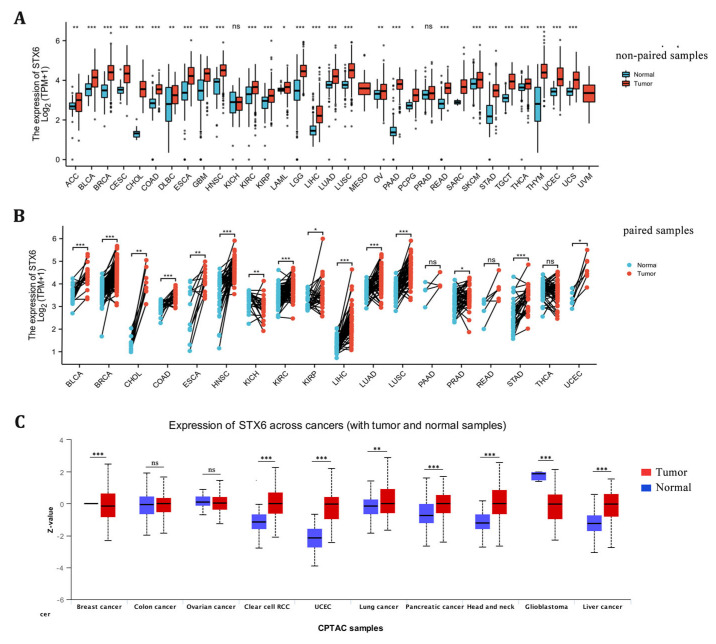
The mRNA and protein levels of STX6 are significantly upregulated in pan-cancer tissues compared to normal tissues. The mRNA level of STX6 is overexpressed in pan-cancer tissues compared to normal samples based on the data from TCGA and GETx databases (**A**,**B**). The same trend was observed at the protein level, especially in clear cell RCC, UCEC, head and neck cancer, and liver cancer (**C**). * *p* < 0.05, ** *p* < 0.01, and *** *p* < 0.001. ns: not significant.

**Figure 2 cancers-15-00027-f002:**
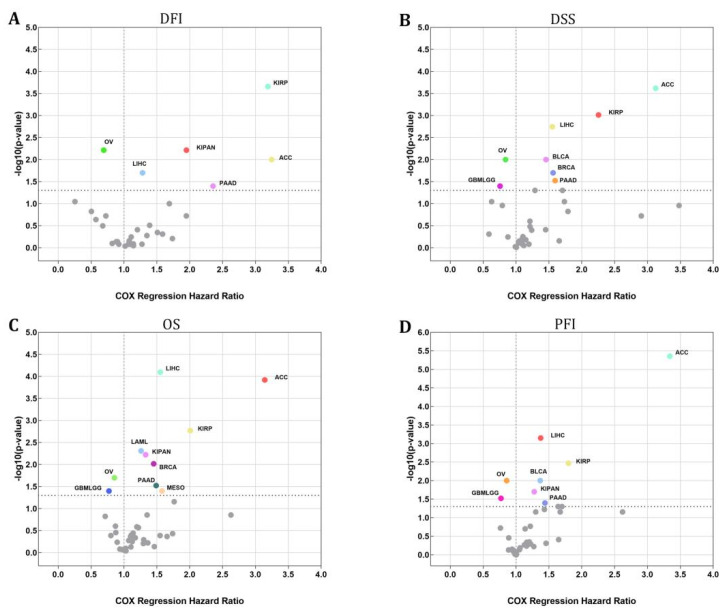
The STX6 level can sufficiently predict the prognosis of patients with cancer. Cox regression analysis showed that patients with high levels of STX6 showed a poor prognosis in DFI, DSS, OS, or PFI, especially patients with LIHC, ACC, PAAD, and KIRP, whereas OV and GBMLGG patients showed the opposite (**A**–**D**).

**Figure 3 cancers-15-00027-f003:**
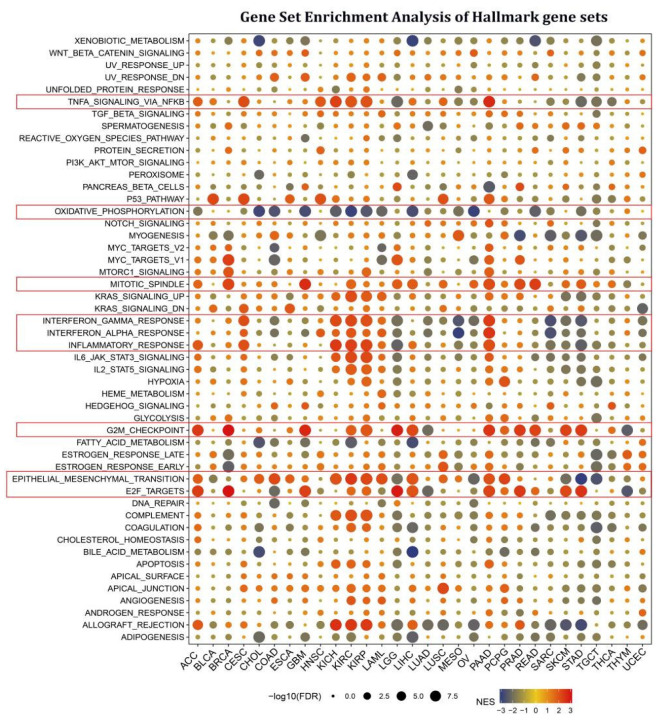
STX6 is involved in EMT, cell cycle, and tumor immunity in pan-cancer via gene set enrichment analysis of hallmark gene sets. STX6 expression was favorably connected to the mitotic spindle, G2M checkpoint, and epithelial–mesenchymal transition. Moreover, STX6 was also linked to TNFA signaling via NFKB, IFN-α, IFN-β, and inflammation, as well as allograft-rejection pathways in kidney cancer (KICH, KIRC, and KIRP), CESC, and PAAD; however, this was not the case for other cancers, including LGG, MESO, OV, SARC, SKCM, and STAD, where STX6 inhibits cellular oxidative phosphorylation.

**Figure 4 cancers-15-00027-f004:**
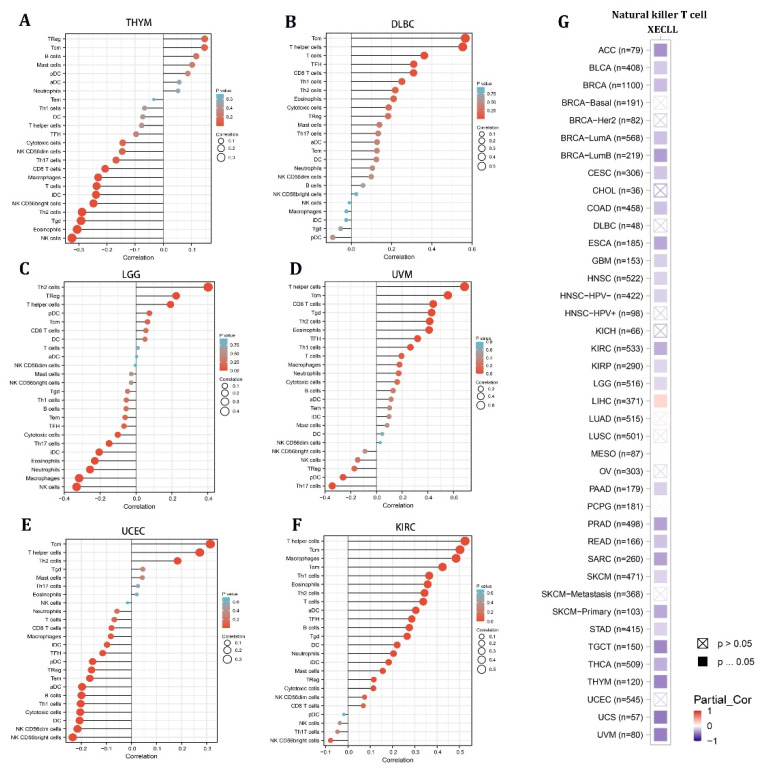
The level of STX6 in cancer tissue can predict the infiltration degree of NK cells and T helper cells. The results of single-sample gene set enrichment analysis (ssGSEA) for STX6 protein showed that malignant tissues with greater STX6 levels exhibited a larger percentage of TCM cells and T helper cells, but lower NK cell levels (**A**–**F**). Furthermore, STX6 was significantly negatively correlated with NKT cells in pan-cancer according to the TIMER database (**G**).

**Figure 5 cancers-15-00027-f005:**
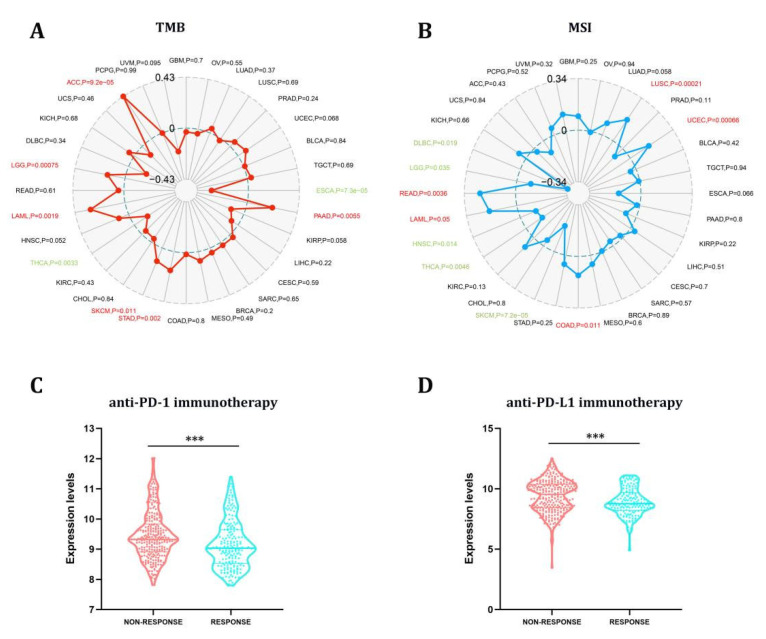
The STX6 expression level can predict the effect of immunotherapy. STX6 expression was favorably connected with TBM in ACC, LGG, LAML, SKCM, STAD, and PAAD while being negatively correlated in THCA and ESCA (**A**). Additionally, STX6 showed a favorable link with READ, LAML, COAD, LUSC, and UCEC, but a negative relationship with the MSI of five different cancers, including DLBC, LGG, HNSC, THCA, and SKCM (**B**). For immunotherapy, STX6 mRNA expression was higher in the non-response group than in the response group independent of treatment with PD-L1 or PD-1 by integrating GEO transcriptome sequencing data on PD-1 and PD-L1 treatment (**C**,**D**). *** *p* < 0.001.

**Figure 6 cancers-15-00027-f006:**
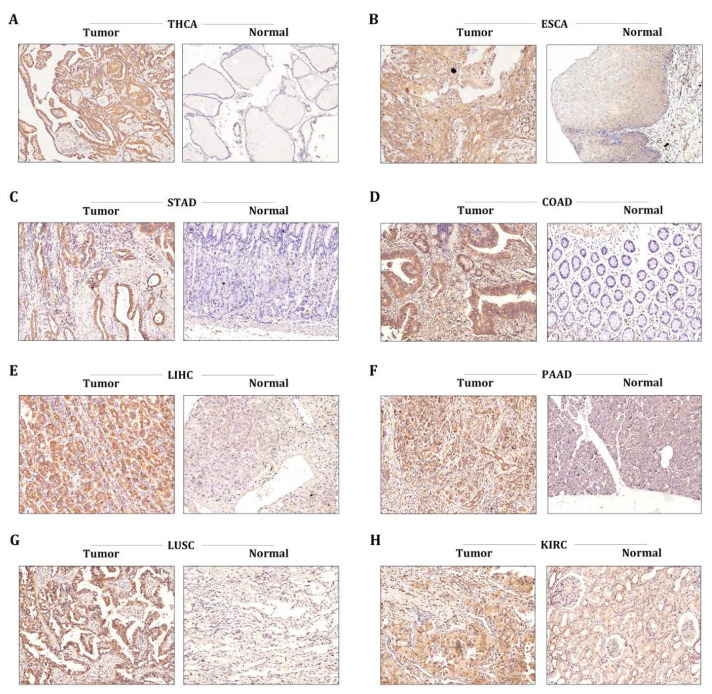
The STX6 level was regulated in multiple cancer clinical specimens. Immunohistochemical analysis showed that the STX6 staining score in tumor tissues was higher than that in normal tissues in all cancer types included in the study (**A**–**H**) (specimens were obtained from author’s institution).

**Figure 7 cancers-15-00027-f007:**
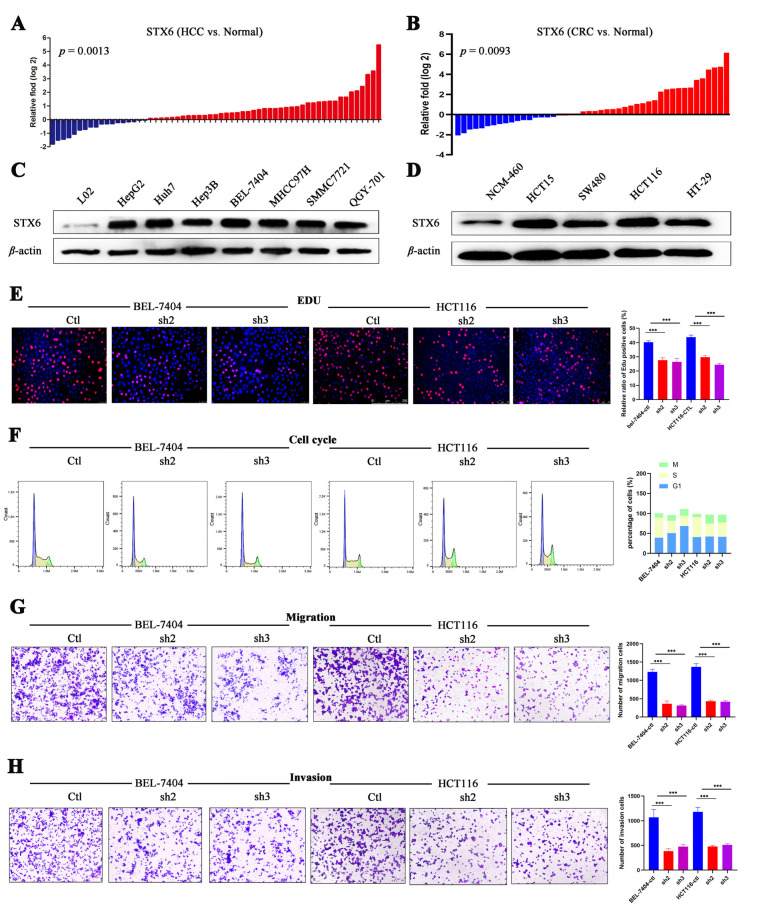
STX6 promotes the proliferation and metastasis of HCC and CRC cells. The mRNA level of STX6 was upregulated in HCC and CRC tissues compared to paired normal samples (**A**,**B**). In HCC and CRC cell lines, the protein level of STX6 was overexpressed compared to normal cell lines (**C**,**D**). Knockdown of STX6 expression inhibited the proliferation (**E**), cell-cycle progression (**F**), and cell invasion and migration (**G**,**H**) of BEL-7404 and HCT-116 cells. *** *p* < 0.001. The whole Western Blot can be found in Appendix A.

**Figure 8 cancers-15-00027-f008:**
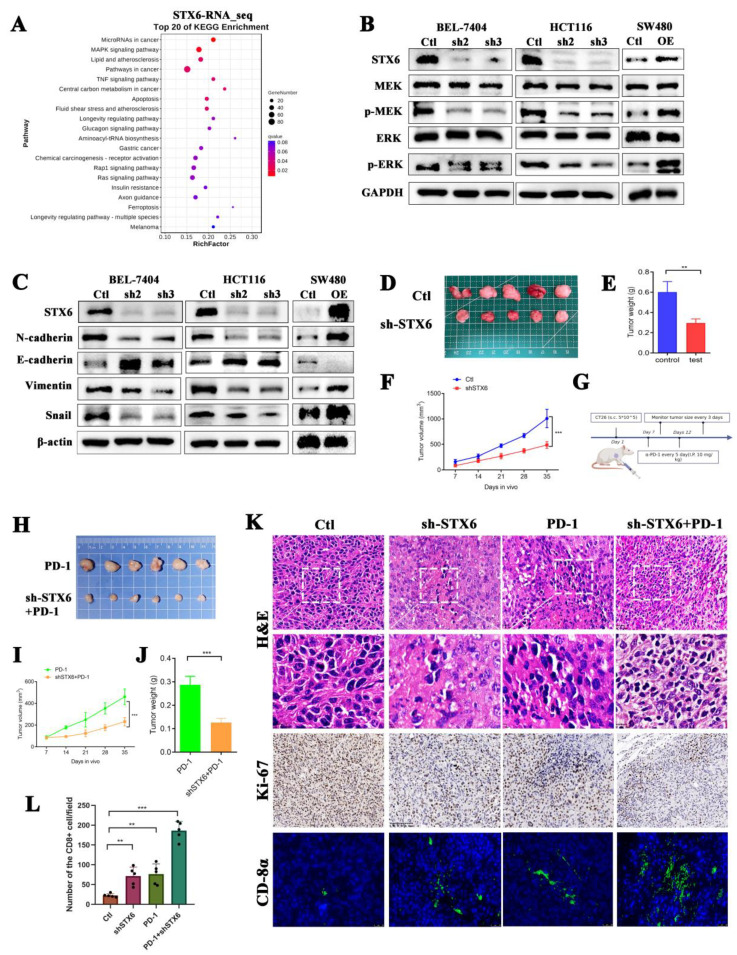
Knockdown of STX6 expression inhibited tumor formation and promoted CD8 cell infiltration in a mouse model. The RNA-seq results about the knockdown of STX6 in BEL-7404 cells showed that STX6 is involved in cancer pathways, microRNA cancer pathways, and MAPK signaling pathways (**A**). Knocking down STX6 levels arrested the MAPK/ERK pathway and EMT pathway in HCC and CRC cells, while overexpression of STX6 promoted the activities of the above two pathways (**B**,**C**). Tumor weight and volume decreased when they were formed subcutaneously by injection of CT26 cells with silencing of STX6 (**D**–**F**). Schematic diagram of CT26 subcutaneous tumorigenesis and anti-PD-1 therapy plan (**G**). Silencing the STX6 level potentiated the anti-PD-1 efficacy in the mouse model (**H**–**J**). The immunofluorescent test (immunohistochemistry) showed that the scores of tumor proliferation-related indices in the STX6 knockdown group or STX6 knockdown combined with anti-PD-1 therapy were lower than those in the control group or anti-PD-1 therapy respectively and promoted CD8 cell infiltration in the mouse model (**K**,**L**). ** *p* < 0.01, *** *p* < 0.001. The whole Western Blot can be found in Appendix A.

**Table 1 cancers-15-00027-t001:** Immune and stromal scores of STX6 in pan-cancer.

Cancer	Immune Score		Stromal Score	
	Spearman_R	*p*-Value	Spearman_R	*p*-Value
ACC	−0.189	0.101	−0.089	0.444
BLCA	−0.094	0.058	−0.174	<0.001
BRCA	−0.110	<0.001	−0.065	0.034
CESC	−0.043	0.466	0.033	0.578
CHOL	−0.022	0.899	0.072	0.676
COAD	0.149	0.012	0.225	<0.001
DLBC	0.504	<0.001	0.251	0.092
ESCA	−0.259	<0.001	−0.046	0.541
GBM	−0.225	0.005	−0.172	0.034
HNSC	−0.149	0.001	−0.003	0.947
KICH	0.205	0.101	0.192	0.126
KIPAN	0.413	<0.001	0.547	<0.001
KIRC	0.243	<0.001	0.447	<0.001
KIRP	0.106	0.075	0.230	<0.001
LAML	0.125	0.130	0.352	<0.001
LGG	−0.335	<0.001	−0.352	<0.001
LIHC	0.065	0.220	0.048	0.361
LUAD	−0.138	0.002	−0.003	0.939
LUSC	−0.238	<0.001	−0.138	0.002
MESO	−0.132	0.228	0.070	0.522
OV	−0.145	0.003	−0.051	0.298
PAAD	0.090	0.235	0.154	0.040
PCPG	−0.088	0.242	0.013	0.867
PRAD	0.019	0.666	0.071	0.115
READ	0.122	0.248	0.160	0.129
SARC	−0.233	<0.001	−0.104	0.094
SKCM	−0.152	0.001	−0.073	0.120
STAD	−0.182	<0.001	−0.184	<0.001
STES	−0.366	<0.001	−0.302	<0.001
TGCT	−0.087	0.323	−0.160	0.066
THCA	−0.254	<0.001	−0.091	0.041
THYM	−0.375	<0.001	−0.251	0.006
UCEC	−0.305	<0.001	−0.216	0.004
UCS	−0.053	0.700	−0.092	0.499
UVM	0.256	0.023	0.208	0.066

## Data Availability

The data presented in this study are available on request from the corresponding author.

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
