# Peer review of "Syntaxin-6, a Reliable Biomarker for Predicting the Prognosis of Patients with Cancer and the Effectiveness of Immunotherapy"

_cancers, 2022, doi:10.3390/cancers15010027_

Round 1
Reviewer 1 Report
This manuscript, original research, written by Dr. Li et al., analyzed the role of syntaxin-6 in cancer, as a potential oncogene, and its relationship with the immune microenvironment. The research included several pan-cancer analyses using publicly available datasets, and later performed in vitro, and in vivo analyses. In summary, syntaxin-6 functions as a promoter of cancer. This manuscript contains a lot of information, and eventually the reader will have the risk of being saturated. The most important parts of the results are the in vivo and in vitro. There, many cell lines were used, and it would be very useful if the authors can write the type of tumor of the cell line. Or at least add the abbreviation list at the end of the manuscript. As I understood, the authors did not use a humanized mice, so I am not sure how the pd-1 inhibitors could affect the microenvironment unless the drug also affects the microenvironment of the mice. Regarding the immunohistochemistry of Figure 6, I was not sure if that was the author’s own or online data. A table or figure with a summary of the most important findings could help the reader.
Specific comments:
(1) Lines 49-50, regarding “Cancer was the leading cause of death for those younger than 70.” Could you please confirm that it is not “more than 70”?
(2) Line 86. Could you please add “The Cancer Genome Atlas and Genotype-Tissue Expression (GTEx) databases”?
(3) Line 89. Could you please confirm the sentence “Less than three samples for cancer were excluded”?
(4) Line 94. Regarding “in two rounds of t-tests.” Could you please confirm the meaning?
(5) Regarding section 2, materials and methods. To ensure reproducibility, could you please add the exact webpages where the data can be accessed, and the exact data files?
(6) Lines 105-106. Could you please add the links for the R packages that were used?
(7) Line 128. Could you please add the catalog number of the secondary antibody “goat antirabbit IgG biotinylated antibody”?
(8) Lines 132-134. Is it possible to have images of the different intensities and percentage of positive cells?
(9) Lines 132-135. Could you please add the formula of the H-score that it is being used?
(10) Line 182. Could you please provide the details of the different cell lines that were used? (CRC and HCC).
(11) Line 212. In this paragraph, it is stated that the cell lines “LIHC” and “CRC” were used. I think it was CRC and HCC according to the Western Blotting paragraph.
(12) Lines 227-233. Regarding the in vivo mouse model. Could you please state how many mice were used? Sintilimab is a PD-1 inhibitor. Does sintilimab react to mice immune cells? Are the mice humanized?
(13) Line 277. It is stated that the STX6 mRNA expression is correlated with copy-number variations. Could you please describe if the copy changes were gains, losses, or CN-LOH?
(14) Lines 286-287. Regarding “STX6 enhanced the results.” Could you please used “associated with favorable survival”? Additionally, is it the overall survival?
(15) Line 296, regarding “led us to pick LIHC, KIRP, or OV”. In Figure 2, many other cancers are shown. Could you please explain why in this sentence only 3 are mentioned?
(16) In the results of Figure 3 and paragraph of lines 307-324, was the GSEA or the ssGSEA technique performed? If it was the “conventional” GSEA, the variable “phenotype” was the gene expression STX6 as “high” vs “low”? Could you please explain with more details what was exactly done?
(17) Regarding Table 1. I am sorry but this data is could be simplified. Do you need to show up to 9 decimals? Could you please underline the values that show a strong association between the cancer type, STX6 and the 2 scores?
(18) Regarding Table 1. LUAD as a 0.94 spearman P score for the stromal score. Why this is not mentioned in line 344? Which is the difference between spearman R and P? Could you please confirm that the results shown between the lines 340-349 are correct?
(19) Line 361, regarding “STX6 promotes immune escape by inhibiting NK cell infiltration and promoting T helper cell differentiation”. Since it is a ssGSEA analysis, is it correct to make this statement? This sentence looks more like an in vitro or in vivo study.
(20) Figure 5 shows the relationship between TMB, MSI, STX6, and anti-pd1 and anti-pd-L1 antibodies. Is this a results from “in silico” analysis? PD-L1 inhibitor was not mentioned previously. Please explain with more details what was done.
(21) Figure 6. Here, I am not sure what kind of immunohistochemistry was done? It is the IHC for STX6 of the TCGA data? Are the cases from the author’s Institution?
(22) Figure 7. Could you please add the details of all the cell lines that were used, including name, and type of tissue/tumor, abbreviation, etc?
(23) In Figure 8K, do the use of PD-1 inhibitor result in increased CD8a+cells, as observed by immunofluorescence? The silencing and pd-1 inhibition seems to increase the CD8+cells. Do CD8+Tc lymphocytes express PD-1? Do the anti-pd-1 inhibitor that it is being used in this experiment affect mice immune cells?
Author Response
Response to Reviewer 1 Comments
- Lines 49-50, regarding “Cancer was the leading cause of death for those younger than 70.” Could you please confirm that it is not “more than 70”?
Response 1: Thanks for the reminder, the correct change has been made here.
- Line 86. Could you please add “The Cancer Genome Atlas and Genotype-Tissue Expression (GTEx) databases”?
Response 2: Thanks for the advice, the link about TCGA and GTEx has been added.
- Line 89. Could you please confirm the sentence “Less than three samples for cancer were excluded”?
Response 3: Thanks for the reminder, the correct change has been made here.
- Line 94. Regarding “in two rounds of t-tests.” Could you please confirm the meaning?
Response 4: Thanks for the reminder, the correct change has been made here.
- Regarding section 2, materials and methods. To ensure reproducibility, could you please add the exact webpages where the data can be accessed, and the exact data files?
Response 5: Thanks for the advice, the webpages for downloading the data have been added.
- Lines 105-106. Could you please add the links for the R packages that were used?
Response 6: Thanks for the advice, the link about R packages has been added.
- Line 128. Could you please add the catalog number of the secondaryantibody “goat antirabbit IgG biotinylated antibody”?
Response 7: Thanks for the advice, the catalog number has been added
- Lines 132-134. Is it possible to have images of the different intensities and percentage of positive cells?
Response 8: Thanks for the comment, but we only included 3 cases of each cancer type for immunohistochemistry in Figure 6. So it is hard to provide images of the different intensities and percentage of positive cells for each cancer type.
- Lines 132-135. Could you please add the formula of the H-scorethat it is being used?
Response 9: Thanks for the advice, the H-score has been added.
- Line 182. Could you please provide the details of the different cell lines that were used? (CRC and HCC).
Response 10: Thanks for the advice; the different cell lines' details have been added.
- Line 212. In this paragraph, it is stated that the cell lines “LIHC” and “CRC” were used. I think it was CRC and HCC according to the Western Blotting paragraph.
Response 11: Thanks for the reminder, the correct change has been made here.
- Lines 227-233. Regarding the in vivo mouse model. Could you please state how many mice were used? Sintilimabis a PD-1 inhibitor. Does sintilimab react to mice immune cells? Are the mice humanized?
Response 12: Thanks for the reminder, the number of mice used in the mouse model has been added and we apologize for our writing mistakes. Instead of using Sintilimab, we used anti-mouse PD-1 for immunotherapy, so we did not use humanized mice either.
- Line 277. It is stated that the STX6 mRNA expression is correlated with copy-number variations. Could you please describe if the copy changes were gains, losses, or CN-LOH?
Response 13: Thanks for the reminder, The types of all copy-number variations are noted in Figure S1.
- Lines 286-287. Regarding “STX6 enhanced the results.” Could you please used “associated with favorable survival”? Additionally, is it the overall survival?
Response 14: Thanks for the advice, the correct change has been made here. Yes, it is
- Line 296, regarding “led us to pick LIHC, KIRP, or OV”. In Figure 2, many other cancers are shown. Could you please explain why in this sentence only 3 are mentioned?
Response 15: Thanks for the reminder, we added the results of ACC and PAAD in Figure S4.
- In the results of Figure 3 and paragraph of lines 307-324, was the GSEA or the ssGSEA technique performed? If it was the “conventional” GSEA, the variable “phenotype” was the gene expression STX6 as “high” vs “low”? Could you please explain with more details what was exactly done?
Response 16: Thanks for the reminder, it was the GSEA analysis about the relationship between STX6 and hallmark gene sets in pan-cancer.
- Regarding Table 1. I am sorry but this data is could be simplified. Do you need to show up to 9 decimals? Could you please underline the values that show a strong association between the cancer type, STX6 and the 2 scores?
Response 17: Thanks for the reminder, the table has been remade.
- Regarding Table 1. LUAD as a 0.94 spearman P score for the stromal score. Why this is not mentioned in line 344? Which is the difference between spearman R and P? Could you please confirm that the results shown between the lines 340-349 are correct?
Response 18: Thanks for the reminder, spearman P is the p-value of spearman correlation analysis.
- Line 361, regarding “STX6 promotes immune escape by inhibiting NK cell infiltration and promoting T helper cell differentiation”. Since it is a ssGSEA analysis, is it correct to make this statement? This sentence looks more like an in vitro or in vivo study.
Response 19: Thanks for the reminder, the correct change has been made here.
- Figure 5 shows the relationship between TMB, MSI, STX6, and anti-pd1 and anti-pd-L1 antibodies. Is this a results from “in silico” analysis? PD-L1 inhibitor was not mentioned previously. Please explain with more details what was done.
Response 20: Thanks for the reminder, the data of PD-1 and PD-L1 were obtained from the GEO database.
- Figure 6. Here, I am not sure what kind of immunohistochemistrywas done? It is the IHC for STX6 of the TCGA data? Are the cases from the author’s Institution?
Response 21: Thanks for the reminder, the correct change has been made here.
- Figure 7. Could you please add the details of all the cell lines that were used, including name, and type of tissue/tumor, abbreviation, etc?
Response 22: Thanks for the reminder, the correct change has been made here.
- In Figure 8K, do the use of PD-1 inhibitor result in increased CD8a+cells, as observed by immunofluorescence? The silencing and pd-1 inhibition seems to increase the CD8+cells. Do CD8+Tc lymphocytes express PD-1? Do the anti-pd-1 inhibitor that it is being used in this experiment affect mice immune cells?
Response 23: Thanks for the reminder, immunofluorescence was conducted to observe the inflation of CD8a+ cells in tumor tissues. Sorry, we did not detect the level of PD-1 in mouse CD8+ Tc lymphocytes after immunotherapy and the effect on mouse immune cells.
Reviewer 2 Report
The manuscript, presented by Li and coworkers, is well written and well organized.
STX-6 protein is very interesting because it is overexpressed in many tumor tissue cells compared with normal cells, and it correlates with the overall survival and prognosis of cancer patients. In fact, STX6 is involved in the tumor microenvironment by going to interfere with VEGF promotion, cell cycle, metastasis process and enhancing the efficacy of PD-L1.
This research involved both tissue and tumor cell experiments that reported the same results, this supports the thesis that STX-6 could be a new biomarker for predicting patient prognosis and immunotherapy efficacy.
In my opinion, this manuscript is suitable for publication in Cancers
Translated with www.DeepL.com/Translator (free version)
Author Response
Thank you for your review